# The Influence of Technological Parameters on the Contrast of Copper Surfaces in the Laser Marking Process

**DOI:** 10.3390/ma18174024

**Published:** 2025-08-28

**Authors:** Lyubomir Lazov, Edmunds Teirumnieks, Emil Yankov, Nikolay Angelov, Risham Singh Ghalot, Plamen Tsankov

**Affiliations:** 1Engineering Center, RTU Rezekne Academy, 4601 Rezekne, Latvia; edmunds.teirumnieks@rtu.lv (E.T.); emil.yankov@rtu.lv (E.Y.); risham-singh.ghalot@edu.rtu.lv (R.S.G.); 2Department of Mathematics, Informatics and Natural Sciences, Technical University of Gabrovo, 5300 Gabrovo, Bulgaria; angelov_np@tugab.bg (N.A.); plamen@tugab.bg (P.T.); 3Center of Competence “Smart Mechatronic, Eco- and Energy-Saving Systems and Technologies”, 5300 Gabrovo, Bulgaria

**Keywords:** laser marking, copper, Yb-doped fiber laser, CuBr MOPA laser, contrast, speed, raster step, effective energy

## Abstract

This study examines the influence of key technological parameters—marking speed, raster step (Δ*x*), pulse duration, power density, and effective energy—on the laser marking of copper using Yb-doped fiber and CuBr MOPA lasers. Two experimental setups were used: the fiber laser, with 100 ns and 200 ns pulses, and the CuBr laser with 30 ns pulses. Marking speed ranged from 10 to 80 mm/s, with raster steps from 3 to 20 µm for the fiber laser and 3 to 27 µm for the CuBr laser. The study compares different pulse durations and evaluates the impact of laser wavelength on the marking process. Optimal effective energy ranges were identified: 17.4–43.1 kJ/cm^2^ for the Yb-doped fiber laser and 9.90–43.1 kJ/cm^2^ for the CuBr laser. The originality of this work lies in its direct comparison of Yb-doped fiber and CuBr MOPA lasers for copper marking, alongside the simultaneous optimization of multiple parameters. The study provides novel guidelines for high-contrast copper marking, a material with known laser-processing challenges. The identified optimal energy ranges and process parameters can significantly improve the efficiency and quality of industrial copper marking applications.

## 1. Introduction

Copper is a semi-precious metal with broad and diverse industrial applications [1]. Its excellent electrical and thermal conductivity, antimicrobial properties, ease of processing, and resistance to environmental influences make it indispensable in various sectors. Research into laser marking of copper surfaces is particularly important for industries such as aeronautics, aircraft construction, medicine, and defense. Laser marking on copper and copper alloys is widely utilized due to its high precision, marking quality, and the ability to produce durable, high-contrast engravings [2,3,4].

However, copper’s reflective nature—especially at infrared wavelengths commonly used in industrial lasers—poses significant challenges for researchers. Its high reflectivity and low absorption hinder efficient energy transfer. Nevertheless, by exceeding specific power density thresholds and employing ultrashort pulses (nanosecond, picosecond, or femtosecond), it is possible to achieve effective material interaction and perform ablation processes suitable for applying serial numbers, matrix codes, and other identification markings on copper components.

Among the various process parameters, marking speed plays a crucial role, as it directly determines the laser’s interaction time with the material. Numerous studies have shown that increasing the marking speed generally reduces contrast, though in some cases this relationship has been reported to be nonlinear [5,6,7]. While slower speeds can enhance marking contrast, they tend to reduce overall process efficiency. Conversely, faster marking improves productivity but may compromise contrast and mark quality [8]. Thus, achieving an optimal balance between contrast and throughput is key to process optimization.

Pulse duration is another critical factor influencing the energy delivered per pulse and the thermal effects induced on the material. Longer pulses typically transfer more energy per pulse, enabling deeper and more intense markings, whereas shorter pulses reduce the heat-affected zone and produce finer, higher-contrast features. Žemaitis [9] investigated pulse durations ranging from 210 femtoseconds to 10 picoseconds and found that shorter pulses yielded superior contrast due to minimized thermal diffusion. In that study, a solid-state laser emitting at 1030 nm and operating at a 64.5 MHz pulse repetition rate was used.

In related work, Husinski et al. [10] explored the effects of pulse duration on the ablation threshold and incubation coefficient for three materials: copper (metal), silicon (semiconductor), and gelatin (biopolymer). Using a Ti:sapphire laser system with pulse durations of 10, 30, 250, and 550 fs, they demonstrated that shorter pulses consistently reduced the ablation threshold across all materials tested.

Mustafa et al. further confirmed that longer pulses (241 ns to 350 fs range) increase energy penetration and ablation depth but degrade surface quality. Their results indicate that shorter pulses result in smoother edges and higher contrast—a trend that is extendable to copper marking. While nanosecond pulses offer deeper penetration, picosecond and femtosecond pulses ensure finer surface resolution, albeit at potentially lower efficiency [11].

Additional studies have focused on the formation of surface microstructures during laser marking of copper, highlighting the influence of speed, fluence, and pulse count [12]. The role of laser parameters in optimizing contrast and surface texture during Nd:YAG laser marking of Data Matrix codes has also been examined [13]. A numerical simulation of laser-induced heating in metals by Tatarinov et al. [14] established correlations between temperature, intensity, and exposure duration, providing foundational insights into mechanisms of melting vs. vaporization-based marking. Other researchers working with MOPA laser systems developed laser systems and investigated their technological parameters on the quality of laser radiation [15,16,17,18,19,20] and their application in laser surface marking and structuring machining [21,22,23].

Although many studies focus on copper, insights from other materials are also relevant. For example, ref. [24] investigates color laser marking of steel using fiber lasers and finds a correlation between marking speed, defocus, and surface coloration. Evaluation methods for matrix code quality under varying process conditions are described in [25].

Overall, the factors influencing laser marking contrast can be grouped into three main categories [26]:Material properties: optical and thermo-physical characteristics;Laser parameters: wavelength, power density, pulse energy, pulse duration, and pulse frequency;Process parameters: marking speed, raster step Δ*x*, defocus, and number of repetitions.

Despite substantial progress, many aspects of laser marking on copper remain underexplored. Each application—depending on component geometry, laser type, and functional requirements—requires tailored optimization through pilot studies.

The main objective of this study is to investigate the influence of key technological parameters—specifically marking speed *v*, raster step Δ*x*, pulse duration *τ*, and effective energy *E_eff_*—on the quality and contrast of laser markings on copper surfaces, with applications in aerospace, medicine, aircraft construction, and defense technologies.

## 2. Material, Equipment, and Methodology

### 2.1. Material

The experiments were performed on Cu-ETP copper samples. It is used for current conductors, rolled products, and high-quality tin-free bronzes for the manufacture of products of cryogenic technology, for the manufacture of wire and rods for automatic welding in an inert gas environment, and submerged arc and gas welding of non-critical structures made of copper, as well as the manufacture of electrodes for welding copper and cast iron [27]. The chemical composition is given in Table 1, and some basic parameters are presented in Table 2. The small amount of impurities in the material is impressive (the copper content is over 99.9%). Copper Cu-ETP has very high coefficients of thermal conductivity and thermal diffusivity.

### 2.2. Laser Systems

Photographs in Figure 1 are the laser technological systems with a fiber laser, and a CuBr laser was used to conduct the experiments. And Table 3 lists the important parameters of the Rofin Power Line F 20 Varia Yb fiber [29] and Copper-Bromide (CuBr) MOPA [30] laser system. Fiber laser F 20 Varia operates in the near infrared region and is a modern laser with very good beam quality and efficiency, whereas the CuBr laser operates in the visible region with wavelengths of λ = 511 and 578 nm but has a small working spot diameter at the focus with a good laser beam quality and high pulse energy.

### 2.3. Laser Microscope

To determine the change of surface structure after laser marking, a used 3D measuring laser microscope OLS5000 (Olympus Corporation, Tokyo, Japan) is equipped with 2 types of optical systems: a color imaging optical system and laser confocal optical system. Through the optical system, the surface microstructure is observed. Through the laser confocal optical system, the surface is scanned, determining the surface roughness and topography of the studied area. For the purpose of surface analysis, an optical magnification of 451× was used, and the studied area has dimensions of 645 µm × 645 µm.

### 2.4. Methodology

To achieve the objective, the following tasks are performed:(1)Influence of speed on contrast in fiber laser marking at 100 ns pulse duration.(2)Influence of speed on contrast in fiber laser marking at 200 ns pulse duration.(3)Influence of raster step on contrast in fiber laser marking.(4)Influence of raster step on contrast in CuBr laser marking.(5)Influence of effective energy on contrast for both lasers.

To conduct the experimental studies, the following steps were taken:

According to the set tasks, four matrices were designed. The three matrices are for the experiments with the fiber laser and contain 6 rows with 8 squares each. The first two of these are to study the dependence of the contrast on the rate for two pulse durations, 100 ns and 200 ns, and 6 powers that are different depending on the pulse duration. The third of them is for the study of the dependence of contrast on the raster step Δ*x* for the faber laser at a pulse duration of 100 ns and 6 speeds. It has 6 rows of 8 squares. The fourth matrix is for the CuBr laser experiments and contains 6 rows of 6 squares. It is for studying the dependence of the contrast on the raster step Δ*x* at a pulse duration of 30 ns and 6 speeds. All experiments are for constant feculence 20 kHz. An example of a designed matrix is given in Figure 2 to carry out tasks 1 and 2. It can be used to obtain the contrast versus velocity for two pulse durations—100 ns (Figure 2a) and 200 ns (Figure 2b).

Copper plates of various sizes are cut. They are 150 mm × 100 mm × 1 mm for the fiber laser experiments and 100 mm × 80 mm × 1 mm for the CuBr laser experiments. All samples before laser processing are cleaned of the thin protective layer of oil by immersion in an ultrasonic bath with isopropanol for 45 min.

Raster marking of squares is performed on each sample according to the assigned tasks and designed matrices. The marked squares were obtained with different technological parameters such as speed, power, pulse duration, raster step, or effective energy. The contrast of the marking is determined for each square.

The methodology for calculating the contrast *k*_x_* is as follows: The visual difference in reflectance between the treated and untreated surfaces is determined using Adobe Photoshop with a specialized algorithm. The contrast *k*_x_* is expressed in relative units or as a percentage (%) on the reference scale. The value of the reflectance *N_f_* of the untreated (unmarked) surface and the value of the reflectance *N_x_* of the laser-marked area are determined. The contrast $k_{x}^{*}$ is calculated using the following expression [31]:(1)$$k_{x}^{*}=\;\frac{N_{f}-N_{x}}{N_{f}}\;\times \;100\%$$
where *N_f_* is the measured value of the reflectivity in the untreated (unmarked) area and *N_x_* is the measured value of the reflectivity in the laser-treated area (marked area).

Five contrast measurements are made for each marked area (square). The root mean square (RMS) error is calculated for each marked square.

For process optimization, the effective energy *E_eff_* value is introduced. The effective energy is related to the laser source and the technological process parameters. It determines the absorbed energy in the impact zone per unit area. The effective energy must be sufficient to melt or vaporize the material, depending on the chosen marking method. It is defined as the multiplication of the linear energy density *E_l_*$=\frac{P}{v}$ and the pulse density linear *I_l_* = $\frac{\upsilon }{v}$. It is given with the expression(2)$$E_{\mathrm{eff}}=\frac{P\upsilon }{v^{2}}$$
where power *P* is in Watts, frequency υ is in Hz, and speed v is in m/s.

The effective energy *E_eff_* has a unit of measurement J/m^2^.

From the obtained experimental results, the graphical dependencies are drawn. The graphs are analyzed, and the obtained results are summarized. A comparison is also made of the results obtained for the two lasers (for the two different wavelengths).

## 3. Results

According to the prepared experimental research project, four matrices were marked on copper plates, which are presented in Figure 3. Matrices 1 (Figure 3a) and 2 (Figure 3b) were marked with a fiber laser, changing two technological parameters—marking speed and power density. The pulse duration was 100 ns for matrix 1 and 200 ns for matrix 2. Matrix 3 (Figure 3c) was marked with a fiber laser with a pulse duration of 100 ns, with the raster step and marking speed being variables. The marking on matrix 4 (Figure 3d) was carried out with a CuBr laser with a pulse duration of 30 ns, again changing the technological parameters—raster step and marking speed.

### 3.1. Influence of Speed on Contrast in Fiber Laser Marking at 100 ns Pulse Duration

Figure 4 shows photographs of fiber-laser-treated areas of copper samples. Images were obtained using a laser microscope. Figure 4a was obtained for power density, *q_s_* = 9.67 kW/mm^2^, and speed, v = 10 mm/s, and Figure 4b, for power density, *q_s_* = 9.67 kW/mm^2^, and speed, v = 70 mm/s. In the first case (Figure 4a), from the photograph taken with the laser microscope, a strongly pronounced melting and boiling of the surface was observed, which leads to higher roughness and high laser contrast (*k** = 67%), while in the second case (Figure 4b), as a result of the high speed, two times less roughness and contrast were obtained with *k** = 23.8%. The influence on roughness depending on the laser processing parameters is presented in studies by the authors Ghalot R.S. et al. [7].

The experimental dependence depicted in Figure 5 illustrates the relationship between contrast and marking speed for a copper plate marked with a fiber laser at a pulse duration of 100 ns and three power densities. The analysis of the graphs reveals several trends and conclusions regarding the influence of speed and power density on contrast.

As the marking speed increases, the contrast decreases across all tested power densities. This decline in contrast is evident throughout the entire speed range, with variations depending on the applied power density. At a lower power density of 4.89 kW/mm^2^, the contrast decreases significantly, dropping from 42% at a speed of 10 mm/s to just 5% at 80 mm/s. Similarly, for a power density of 6.86 kW/mm^2^, the contrast falls from 53% at 10 mm/s to 9% at 80 mm/s. At the highest power density of 9.67 kW/mm^2^, the initial contrast is the highest, starting at 67% at 10 mm/s, but it also decreases sharply to 13% at 80 mm/s.

The nature of the contrast curves reveals different trends across varying speed intervals. Between 10 mm/s and 40 mm/s, the curves show a gradual decrease, indicating a slower rate of contrast reduction. Additionally, in the higher speed range, from 40 mm/s to 80 mm/s, the curves become steeper, reflecting a faster decline in contrast. This sharp drop in the higher speed range suggests that the marking process becomes less effective at higher speeds, regardless of the power density used.

In summary, the results highlight the significant influence of marking speed and power density on contrast when using a fiber laser at a pulse duration of 100 ns. Lower speeds and higher power densities result in higher contrast, making them more suitable for applications requiring clear and precise markings. Higher speeds and lower power densities, however, are more appropriate for laser markings that will be interpreted using specialized readers. The findings emphasize the importance of optimizing these parameters to achieve the desired contrast levels in industrial laser marking processes.

### 3.2. Influence of Speed on Contrast in Fiber Laser Marking at 200 ns Pulse Duration

The graph in Figure 6 illustrates the relationship between contrast and speed for copper plates marked using a fiber laser with a pulse duration of 200 ns. Three power densities ns—10.4 kW/mm^2^, 14.6 kW/mm^2^, and 20.2 kW/mm^2^—were studied to evaluate their impact on marking contrast at different speeds. The results demonstrate the behavior of marking contrast as a function of speed and power density within specific ranges.

At all tested power density values, the contrast decreases nonlinearly with increasing speed. This trend is attributed to the reduced interaction time between the laser and the material at higher speeds, leading to lower energy absorption by the material:At the lowest power density of 10.4 kW/mm^2^, the contrast decreases significantly, from 68% at 10 mm/s to 40% at 80 mm/s—a 1.7-fold reduction. The steep decline indicates that, at this power level, higher speeds fail to provide sufficient energy for creating high-contrast markings, making it less suitable for applications requiring both high speed and high-quality markings.At a power density of 14.6 kW/mm^2^, the contrast decreases from 74% at 10 mm/s to 50% at 80 mm/s, representing a reduction of about 1.48 times. The decrease is less pronounced compared to 10.4 kW/mm^2^, suggesting that the increased power partially compensates for the reduced interaction time at higher speeds, maintaining better marking quality.At the highest power density of 20.2 kW/mm^2^, the contrast decreases from 79% at 10 mm/s to 63% at 80 mm/s, a reduction of only 1.25 times. The minimal loss of contrast indicates that higher power levels are more effective in maintaining marking quality over a broader range of speeds, making this setting ideal for high-speed marking applications.

### 3.3. Comparison of Fiber Laser Marking Results with Pulse Durations of 100 ns and 200 ns

In this study, the influence of pulse duration was analyzed for two cases: 100 ns and 200 ns. Due to the design characteristics of the fiber laser, the pulse energy differs by a factor of two—500 μJ in the first experiment and 1000 μJ in the second. Varying the marking speed from 10 mm/s to 80 mm/s in both experiments results in an 8-fold change in the exposure time during the interaction between the laser radiation and the material. Analysis of the graphical dependencies shows that, in Figure 7, the rate of contrast reduction for the blue curve (100 ns) is twice as high as that for the green curve (200 ns). This behavior is attributed to the combined effect of pulse energy and exposure time on the outcome of the technological process.

Despite variations in contrast, high-contrast laser markings are consistently achieved within the studied range, making this technique highly suitable for industrial marking applications. For higher-speed applications, higher power density values (e.g., 20.2 kW/mm^2^) are recommended to maintain the necessary contrast at faster scanning speeds. Conversely, lower power density values are sufficient for applications with slower speeds, though this is accompanied by reduced production efficiency.

The RMS error in determining contrast ranges from 0.60% to 1.35%, confirming the high accuracy and reliability of the results.

### 3.4. Influence of Raster Step on Contrast in Fiber Laser Marking

The graphs in Figure 8 illustrate the relationship between contrast (*k∗*) and raster step (Δ*x*) for copper samples marked using a fiber laser at three different speeds: 15 mm/s (blue line), 45 mm/s (green line), and 90 mm/s (red line). The analysis of the results demonstrates the influence of raster step and scanning speed on the quality of the laser marking. 

For all three speeds, the contrast decreases non-linearly as the raster step increases. This behavior reflects how larger raster steps reduce the overlap of the laser beam, decreasing the number of repetitions and absorbed energy, and consequently, the contrast. The graphs indicate that the rate of contrast reduction becomes less pronounced as the raster step increases.

At the slowest speed of 15 mm/s, the contrast decreases from 83% to 65% as the raster step increases from 3 µm to 20 µm. This represents a 1.22-fold reduction in contrast. The relatively gradual decrease suggests that this scanning speed allows for higher energy density and better marking quality, even with larger raster steps.

At a speed of 45 mm/s, the contrast decreases from 82% to 62% over the same range of raster steps, corresponding to a 1.27-fold reduction. This slightly steeper decline compared to 15 mm/s suggests that the reduced interaction time between the laser and the material has a more significant impact on contrast and marking quality as the raster step increases.

At the fastest speed of 90 mm/s, the contrast decreases from 80% to 57% as the raster step increases from 3 µm to 20 µm, representing a 1.44-fold reduction. The more pronounced decrease at this speed reflects shorter interaction times and lower energy absorption, which adversely affect the marking contrast.

Despite the observed reduction in contrast at all speeds and raster steps, the data show that high-contrast laser markings are still achieved within the studied range. This makes fiber laser marking suitable for industrial applications requiring durable and visually distinct markings.

### 3.5. Influence of Raster Step on Contrast in CuBr Laser Marking

The graphs in Figure 9 show the relationship between marking contrast and raster step for three different speeds: 15 mm/s (blue line), 45 mm/s (green line), and 90 mm/s (red line). Their analysis reveals the trends and variations in contrast at the raster step and speed values suitable for industrial production.

For all three speeds, contrast decreases non-linearly as the raster step increases. This trend emphasizes that larger raster steps reduce beam overlap, resulting in less absorbed energy and thinner molten or oxidized layers for the largest raster steps. Consequently, a sharper contrast drop is observed.

At a scanning speed of 15 mm/s, the marking contrast decreases from 89% to 63% as the raster step increases from 3 µm to 27 µm. This represents a 1.29-fold reduction in contrast within the examined range. The relatively high initial contrast and slower reduction at this speed indicate that lower scanning speeds allow for greater energy absorption and better marking quality, even as the raster step increases.

At a scanning speed of 45 mm/s, contrast decreases from 80% to 53% over the same range of raster steps (3 µm to 27 µm), corresponding to a 1.34-fold reduction. The faster drop compared to 15 mm/s suggests that increasing the scanning speed reduces the interaction time between the laser and the material, making the markings less distinct at larger raster steps. However, the contrast remains sufficiently high for industrial applications across the entire working range of raster steps.

At the highest scanning speed of 90 mm/s, contrast drops from 70% to 44% as the raster step increases from 3 µm to 27 µm, representing a 1.7-fold reduction. The more pronounced contrast decrease at this speed reflects less absorbed energy by the material due to the rapid movement of the laser beam. For raster steps larger than 22 µm, the quality of the resulting marking is unsuitable for visual recognition.

The results also demonstrate the effectiveness of the CuBr laser in achieving high levels of contrast with relatively lower radiation power compared to fiber lasers. This efficiency is particularly beneficial for applications requiring precise markings with minimal energy consumption.

### 3.6. Influence of Effective Energy on Contrast for Both Lasers

Figure 10 illustrates the relationship between marking contrast k∗ and effective energy (*E_eff_*) for copper samples marked using two types of lasers: a fiber laser (blue curve) and a CuBr laser (green curve). Effective energy is a composite parameter that accounts for the influence of power and process-related factors, such as speed and raster step. In the experiments, the effective energy ranged from 3.03 kJ/cm^2^ to 43.1 kJ/cm^2^ for the fiber laser and from 1.36 kJ/cm^2^ to 43.1 kJ/cm^2^ for the CuBr laser.

The following conclusions can be made from the two graphical dependencies presented:A nonlinear dependence of the marking contrast on the effective energy is obtained for both lasers.For the effective energy intervals from 3.03 kJ/cm^2^ to 15.52 kJ/cm^2^ for the fiber laser and from 1.36 kJ/cm^2^ to 15.52 kJ/cm^2^ for the CuBr laser, the curves are very steep, which indicates a rapid increase in contrast in these intervals.For the effective energy interval of 15.52 kJ/cm^2^ to 43.1 kJ/cm^2^, the rate of contrast increase is very slow for both lasers, suggesting a saturation effect where additional energy has a negligible effect on contrast. In this interval, marking is mainly by melting, with only minor increases in marking depth.Based on the experimental studies, two optimal intervals for efective energy have been determined for both lasers: from 17.4 kJ/cm^2^ to 43.1 kJ/cm^2^ for the fiber laser and from 9.90 kJ/cm^2^ to 43.1 kJ/cm^2^ for the CuBr laser. At equivalent effective energy values, the CuBr laser exhibits higher marking contrast than the fiber laser. This difference is likely due to the material’s higher absorption capacity for the wavelengths of the CuBr laser (λ = 511 nm and 578 nm) compared to that of the fiber laser (λ = 1030 nm), as confirmed by the studies of M. Hummel [32].

## 4. Conclusions

This study examined the impact of key technological parameters—speed (*V*), raster step (Δ*x*), power density (*q_s_*), pulse duration (*τ*), and effective energy (*E_eff_*)—on the laser marking process of copper using Yb-doped fiber and CuBr MOPA lasers. Experimental dependencies were analyzed, focusing on the relationship between contrast and speed (*k* = k**(*V*)) for different power densities and pulse durations, as well as contrast with raster step (*k* = k**(Δ*x*)) and effective energy (*k* = k**(*E_eff_*)) for both lasers. A comparative analysis of the two wavelengths (1030 nm for the fiber laser and 578 and 511 nm for the CuBr laser) was also performed.

Key findings include the following:Contrast decreases nonlinearly with increasing speed for all pulse durations and power densities. For example, at 100 ns and 9.67 kW/mm^2^, contrast dropped from 67% to 13% as speed increased from 10 mm/s to 80 mm/s;As the raster step increases, contrast also decreases for both lasers. At 45 mm/s, the contrast with the fiber laser decreased from 82% to 62% as Δ*x* ranged from 3 µm to 20 µm, while with the CuBr laser, it dropped from 80% to 53% as Δ*x* increased from 3 µm to 27 µm;The CuBr laser requires less effective energy to achieve the same contrast as the fiber laser due to its higher absorption capacity. However, the fiber laser’s higher efficiency makes it more suitable for copper marking;These results offer valuable insights for laser system operators, providing a foundation for creating technological tables with key parameters for marking various products. The methodology can be extended to other materials and laser types.

This study represents the first direct comparative analysis of Yb-doped fiber and CuBr MOPA lasers for copper marking, employing a multiparameter optimization approach. It defines optimal effective energy ranges: 17.4–43.1 kJ/cm^2^ for the fiber laser and 9.90–43.1 kJ/cm^2^ for the CuBr laser, providing practical, industry-relevant guidelines. This framework can be applied to future research on high-contrast laser marking.

## Figures and Tables

**Figure 1 materials-18-04024-f001:**
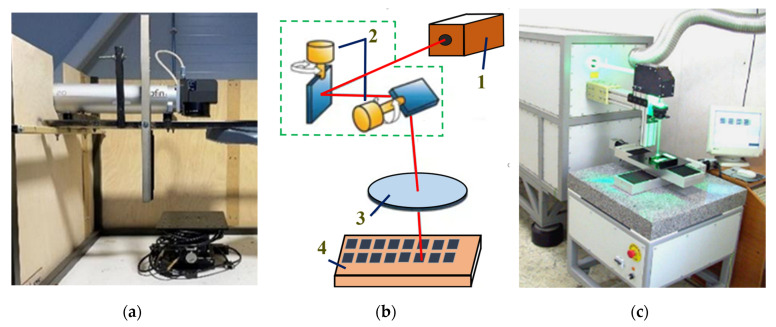
Views of laser systems with (**a**) Rofin Power Line F 20 Varia Yb-doped fibre laser [Latvia]; (**b**) block diagram of laser systems 1—laser source; 2—Galvo motor in X and Y; 3—focusing lens; and 4—sample; (**c**) CuBr MOPA laser [Bulgaria].

**Figure 2 materials-18-04024-f002:**
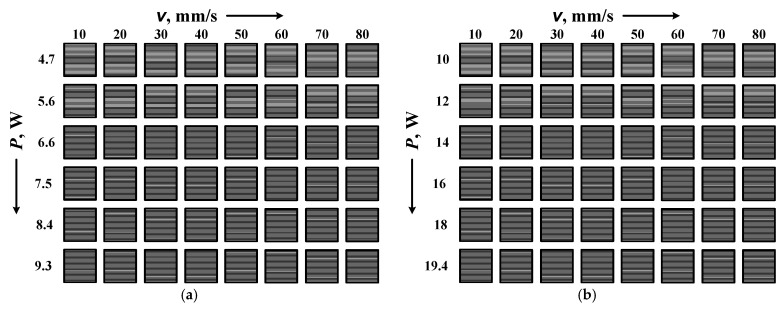
Fiber laser marking matrix designs at pulse durations, (**a**) 100 ns and (**b**) 200 ns, and different speeds and powers.

**Figure 3 materials-18-04024-f003:**
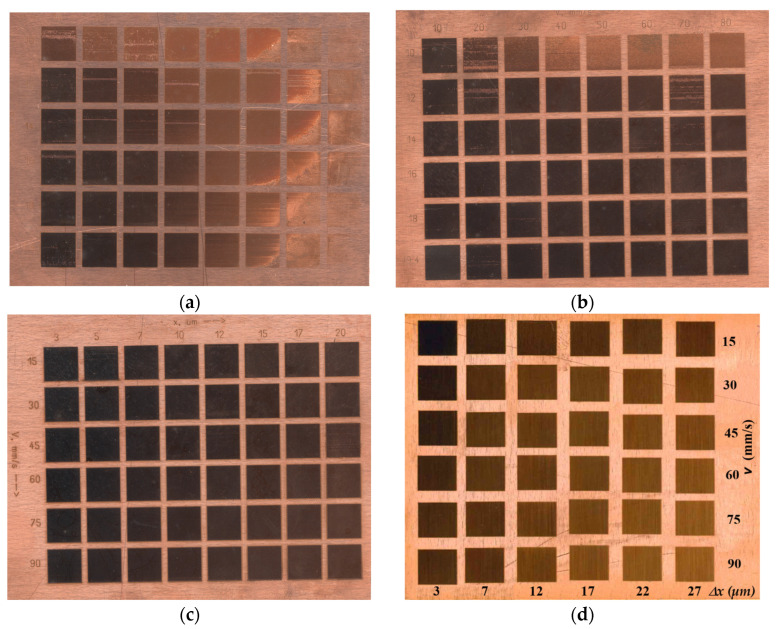
Copper matrices marked with different lasers: (**a**–**c**) Matrices 1–3 marked with a fiber laser; (**d**) Matrice 4 marked with a CuBr laser.

**Figure 4 materials-18-04024-f004:**
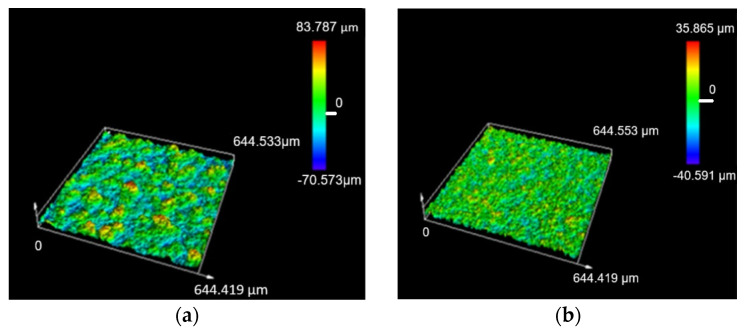
Images of laser-marked areas on a copper sample at a power density of *q_s_* = 9.67 kW/mm^2^ and speeds of v: (**a**) 10 mm/s, *R_z_* = 150 ± 4 μm; (**b**) 70 mm/s, *R_z_* = 75 ± 4 μm.

**Figure 5 materials-18-04024-f005:**
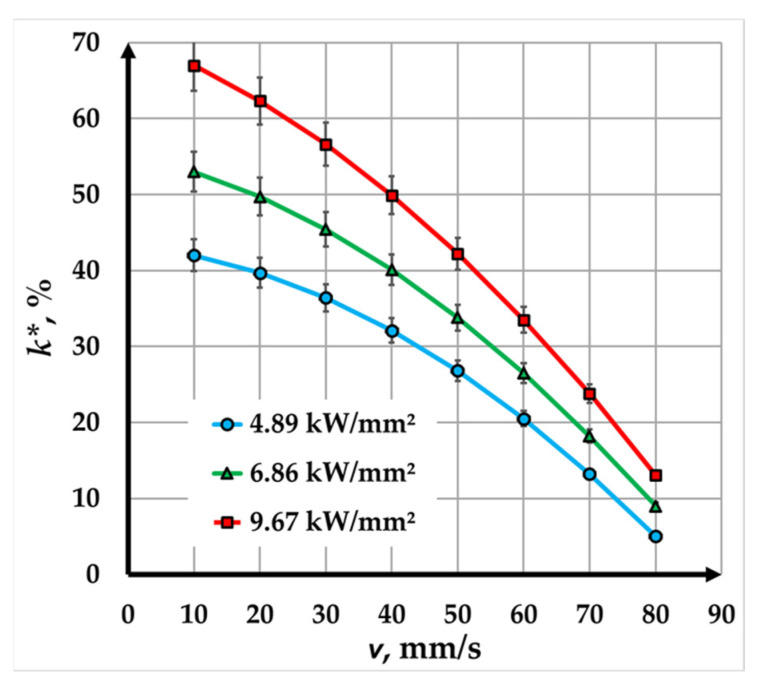
Graphics of the dependence of contrast on speed for marking a Cu plate with a fiber laser with a pulse duration of 100 ns for three power densities: blue color—*q_s1_* = 4.89 kW/mm^2^; green color—*q_s2_* = 6.86 kW/mm^2^; and red color—*q_s__3_* = 9.67 kW/mm^2^.

**Figure 6 materials-18-04024-f006:**
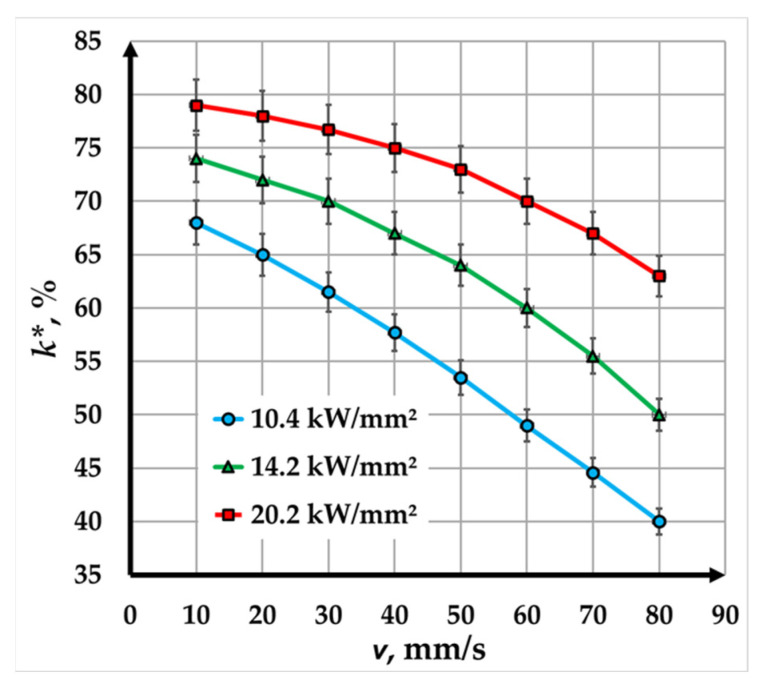
Graphics of the dependence of contrast on speed for marking a Cu plate with a fiber laser with a pulse duration of 200 ns for three power densities: blue color—*q_s1_* = 10.4 kW/mm^2^; green color—*q_s2_* = 14.2 kW/mm^2^; and red color—*q_s__3_* = 20.2 kW/mm^2^.

**Figure 7 materials-18-04024-f007:**
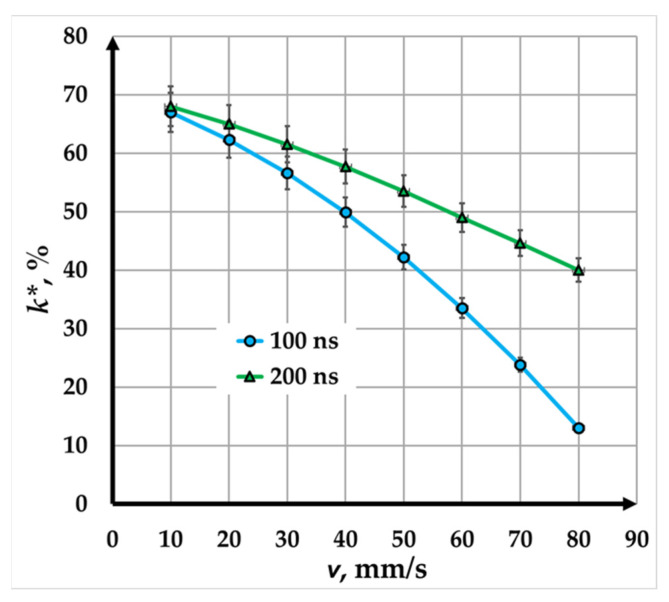
Graphics of the dependence of contrast on speed for marking a Cu plate with a fiber laser with a pulse duration of the following: blue—100 ns; green—200 ns.

**Figure 8 materials-18-04024-f008:**
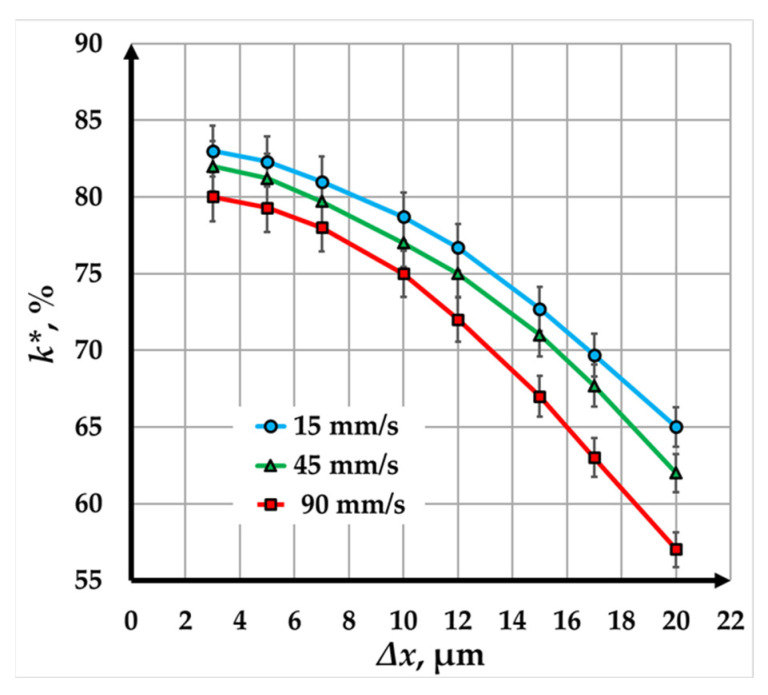
Graphics of the dependence of contrast on raster step for marking a Cu plate with a fiber laser with a pulse duration of 100 ns for three speeds: blue color—15 mm/s; green color—45 mm/s; and red color—90 mm/s.

**Figure 9 materials-18-04024-f009:**
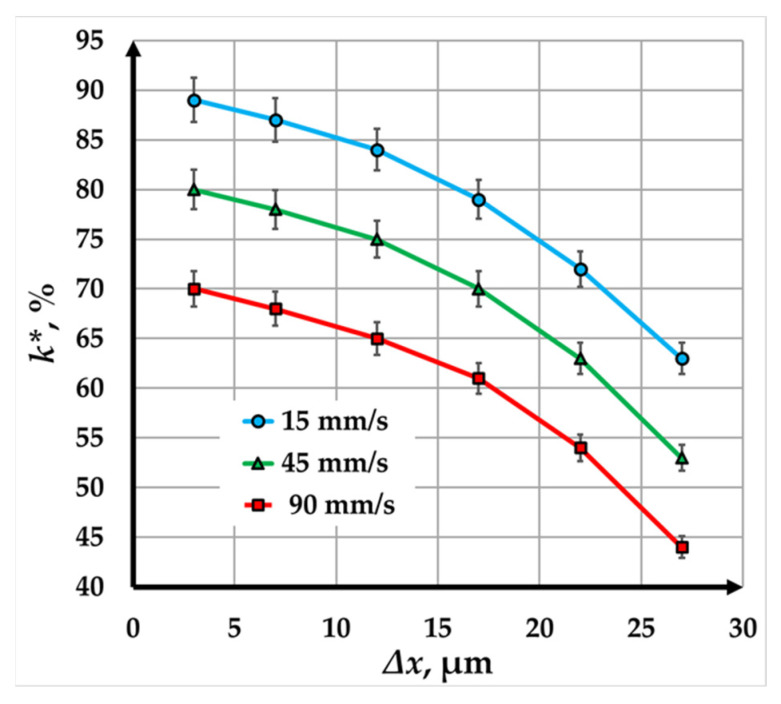
Graphics of the dependence of contrast on the raster step for marking a Cu plate with a CuBr laser with a pulse duration of 30 ns for three speeds: blue color—15 mm/s; green color—45 mm/s; and red color—90 mm/s.

**Figure 10 materials-18-04024-f010:**
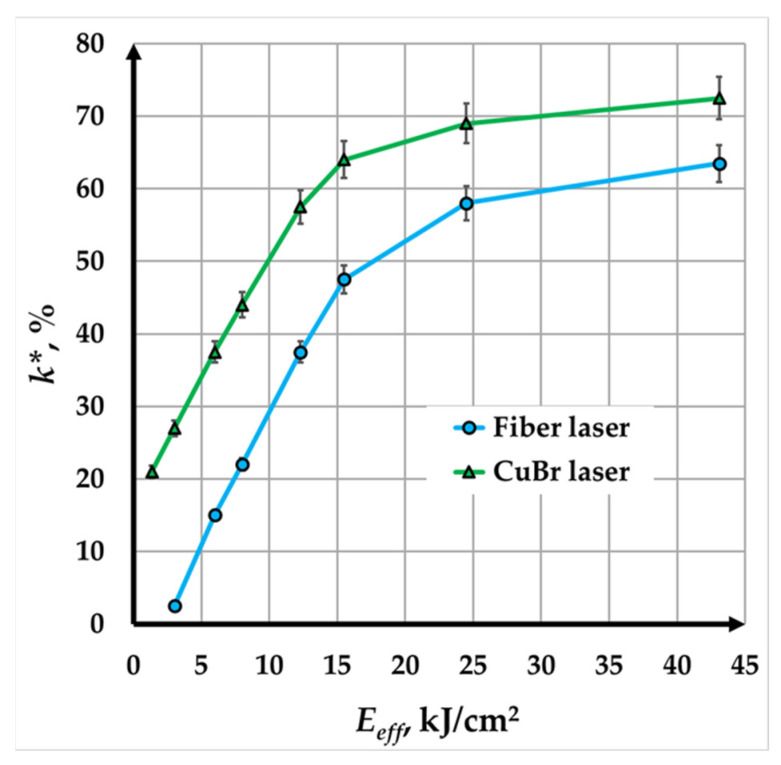
Graphics of the dependence of the contrast on the effective energy for both lasers.

**Table 1 materials-18-04024-t001:** Chemical composition of copper Cu-ETP.

Chemical Element	Content, %
Fe	0.005
Ni	0.002
S	0.004
As	0.002
Pb	0.005
Zn	0.004
O	0.05
Sn	0.002
Bi	0.001
Cu	Balance

**Table 2 materials-18-04024-t002:** Main physical parameters of copper Cu-ETP [28].

Parameter	Value
Coefficients of thermal conductivity *k*, W/(m.K)	400
Density *ρ*, g/cm^3^	8.96
Specific heat capacity *c*, J/(kg.K)	385
Coefficients of thermal diffusivity *a*, m^2^/s	1.16 × 10^−4^

**Table 3 materials-18-04024-t003:** Parameters of laser systems used in the research.

Laser Parameter	Fiber Laser	CuBr Laser
Wavelength *λ*, nm	1030	511 and 578
Power *P*, W	20	10
Diameter in focus *d*, µm	30	30
Frequency *ν*, kHz	20–200	20
Pulse duration *τ*, ns	4–200	30
Pulse energy *E_p_*, mJ	0.1–1.0	0.5
Beam quality *M*^2^	1.1	1.5
Efficiency, %	40	20

## Data Availability

The original contributions presented in this study are included in the article. Further inquiries can be directed to the corresponding author.

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
