# Peer review of "The Influence of Technological Parameters on the Contrast of Copper Surfaces in the Laser Marking Process"

_materials, 2025, doi:10.3390/ma18174024_

Round 1

Reviewer 1 Report

Comments and Suggestions for Authors

The authors have performed a good experimental work and the manuscript has a moderately high novelty. It contributes to the understanding of the influence of process parameters on laser marking of copper, especially through the comparison of Yb fiber and CuBr lasers, which is rare in the literature. Determining the optimal ranges adds value for academic research and industrial applications. The proposed study can serve as a basis for further developments in the field of laser processing of copper. The work is not without shortcomings and requires corrections and can be published after making appropriate changes.

1) In the literature review, the authors do not consider other studies that used CuBr MOPA laser for marking copper or its alloys, as a result of which the novelty of the work remains undisclosed. The authors should formulate the novelty of their work more clearly, in addition to listing the process parameters of laser processing.

2) What is the purpose of the handwritten notes in Figure 3(d)? These notes are unclear. Either remove them or make them understandable.

3) In the caption to Fig. 4, there is a typo in the dimension (line 200): kW/mm^2.

4) Did the authors evaluate the roughness of the substrate surface after processing with a CuBr MOPA laser? It is worthwhile to provide the results obtained for comparison with a fiber laser; this information may be useful for specialists in various fields and will increase the scientific and practical value of the work.

5) In Fig. 5-10, the measurement error should be indicated.

6) On p. 12 (lines 363-366), the authors made an assumption: «At equivalent effective energy values, the CuBr laser produces higher marking con-363 trast than the fiber laser. This difference is probably due to the higher absorption capacity 364 of the material for the CuBr laser wavelengths (λ = 511 nm & 578 nm) compared to that for 365 the fiber laser wavelength (λ = 1030 nm).» It is necessary to confirm it experimentally, for example, using absorption spectroscopy.

7) In the abstract and conclusion, it is necessary to emphasize what the originality of the work consists of.

Author Response

Comments 1: In the literature review, the authors do not consider other studies that used CuBr MOPA laser for marking copper or its alloys, as a result of which the novelty of the work remains undisclosed. The authors should formulate the novelty of their work more clearly, in addition to listing the process parameters of laser processing.

Response 1: Thank you for pointing this out. Eleven new literature sources have been added to the introduction, presenting studies that used the CuBr MOPA laser for marking metals. In the summary and conclusion, we highlight the distinctive originality of the work and the newly investigated aspects of the laser marking process on copper samples and products. (lines 76-79)

Comments 2: What is the purpose of the handwritten notes in Figure 3(d)? These notes are unclear. Either remove them or make them understandable.

Response 2: In Figure 3(d), the information has been re-plotted, making it more understandable and clearer for readers (lines 193 -194)

Comments 3:  In the caption to Fig. 4, there is a typo in the dimension (line 200): kW/mm^2.

Response 3: Thank you for the noted remark on line (200). The necessary correction has been made - kW/mm2 . (line 209)

Comments 4: Did the authors evaluate the roughness of the substrate surface after processing with a CuBr MOPA laser? It is worthwhile to provide the results obtained for comparison with a fiber laser; this information may be useful for specialists in various fields and will increase the scientific and practical value of the work.

Response 4: Results of the measured roughness of the substrate and after processing with a CuBr MOPA laser were investigated as a function of speed, power and raster pitch and published in our article Coatings 20 November 2023 [23] [ https://doi.org/10.3390/coatings13111970] “Investigation of the Change in Roughness and Microhardness during Laser Surface Texturing of Copper Samples by Changing the Process Parameters”. (lines 203 -205)

Comments 5: In Fig. 5-10, the measurement error should be indicated

Response5: Thank you for your suggestion. In Fig. 5-10 we have made the necessary additions and the accuracy of each measurement is indicated. (lines 232, 260,274, 308, 345 and 356)

Comments 6: On p. 12 (lines 363-366), the authors made an assumption: «At equivalent effective energy values, the CuBr laser produces higher marking con-363 trast than the fiber laser. This difference is probably due to the higher absorption capacity 364 of the material for the CuBr laser wavelengths (λ = 511 nm & 578 nm) compared to that for 365 the fiber laser wavelength (λ = 1030 nm).» It is necessary to confirm it experimentally, for example, using absorption spectroscopy.

Response 6: Thanks for the remark. A reference by M. Hummel [30] has been added, in which the absorption of laser light in materials is studied depending on the wavelengths of the laser. The following addition has been made to the text (line 371), which provides better clarity: “This difference is probably due to the higher absorption capacity of the material for the wavelengths of the CuBr laser (λ = 511 nm and 578 nm) compared to that of the fiber laser (λ = 1030 nm), as confirmed by the studies of M. Hummel [30]”. (lines 369 - 375)

Comments 7: In the abstract and conclusion, it is necessary to emphasize what the originality of the work consists of.

Response 7: Thank you for your comment. We have made the necessary additions to the summary and conclusion of the publication so that the originality of our research is clearly visible. (lines 19 – 24; lines 377 - 403)

Reviewer 2 Report

Comments and Suggestions for Authors

     Based on the content of the paper, the following revisions are recommended:

  1. The unit of formula (2) is incorrect. Please check.
  2. The material parameters are incorrect (Table 2). Please check the density of Cu-ETP.
  3. When comparing 100 ns and 200 ns pulses in Section 3.3, the power densities of the two groups are different. It is recommended to add a comparison under the same power density or explain why the parameters are not aligned.
  4. It is not explained why CuBr laser (visible light wavelength) has higher contrast at the same effective energy.
  5. Table 1 (Cu-ETP composition) cites "[20]", but Table 2 (physical properties) has no citation source. It is recommended to standardize the references and data citations.
  6. The conclusion states that fiber laser is more suitable for marking copper than CuBr laser (Section 13), but Section 3.6 points out that CuBr has higher contrast at the same energy, and the efficiency of fiber laser is only twice that of CuBr (Table 3: 40% vs. 20%). Please check the description here.

Author Response

Comments 1: The unit of formula (2) is incorrect. Please check.

Response 1: Thank you for the inaccuracy identified in formula (2). All units are now corrected to the Si system (W, Hz, m/s, J/m2). (lines 179 - 180)

Comments 2: The material parameters are incorrect (Table 2). Please check the density of Cu-ETP.

Response 2: Thank you for your comment. We have made the necessary correction - the correct Density ρ = 8.96 g/cm3. (line 110)

Comments 3:  When comparing 100 ns and 200 ns pulses in Section 3.3, the power densities of the two groups are different. It is recommended to add a comparison under the same power density or explain why the parameters are not aligned.

Response 3: Thanks for the recommendation. The necessary correction was made and inserted in the text (line 264)."In this study, the influence of pulse duration was analyzed for two cases: 100 ns and 200 ns. Due to the design characteristics of the fiber laser, the pulse energy differs by a factor of two - 500 μJ in the first experiment and 1000 μJ in the second. Varying the marking speed from 10 mm/s to 80 mm/s in both experiments results in an eight-fold change in the exposure time during the interaction between the laser radiation and the material. Analysis of the graphical dependencies shows that, in Figure 7, the rate of contrast reduction for the blue curve (100 ns) is twice as high as that for the green curve (200 ns). This behavior is attributed to the combined effect of pulse energy and exposure time on the outcome of the technological process." (lines 265-273)

Comments 4: It is not explained why CuBr laser (visible light wavelength) has higher contrast at the same effective energy.

Response 4: Thanks for the remark. A reference by M. Hummel [33] has been added, in which the absorption of laser light in materials is studied depending on the wavelengths of the laser. The following addition has been made to the text (line 371), which provides better clarity: “This difference is probably due to the higher absorption capacity of the material for the wavelengths of the CuBr laser (λ = 511 nm and 578 nm) compared to that of the fiber laser (λ = 1030 nm), as confirmed by the studies of M. Hummel [33]”. (lines 372-375)

Comments 5: Table 1 (Cu-ETP composition) cites "[20]", but Table 2 (physical properties) has no citation source. It is recommended to standardize the references and data citations.

Response5: Thank you for the remark. A typographical error was made in Table 1, which has been removed. A literature source [22] has been added to Table 2, from which the Physical characteristics of Cu are taken and cited. (line 110)

Comments 6: The conclusion states that fiber laser is more suitable for marking copper than CuBr laser (Section 13), but Section 3.6 points out that CuBr has higher contrast at the same energy, and the efficiency of fiber laser is only twice that of CuBr (Table 3: 40% vs. 20%). Please check the description here.

Response 6: Thank you for your comment. The conclusion has been substantially revised to take your recommendations into account (See Conclusion from lines 377 - 403)

Reviewer 3 Report

Comments and Suggestions for Authors

The manuscript studies how marking speed, raster step (Δx), pulse duration, power density, and a derived effective energy affect laser‑marking contrast on Cu‑ETP, using a 1030 nm Yb‑fiber system and a CuBr MOPA laser. Four experimental matrices are presented and contrast is quantified for a wide parameter space. The topic is timely and of practical relevance to industrial part identification. However, several core issues must be corrected. I recommend major revision.

  1. In Table 2, the density is given as ρ = 0.002 kg/m³, while the specific heat capacity is reported as c = 8940 J/(kg·K). These values are not physically plausible for copper and appear to be swapped (and/or units misassigned). Please correct the numbers and units.
  2. Definition of “effective energy” omits hatch spacing and is internally inconsistent with your text. You define as the absorbed energy per unit area, derived from linear energy density and linear pulse density. However, areal energy input for raster marking also depends on the line‑to‑line spacing Δx, as you rightly noted further E_eff “accounts for … speed and raster step,” but Δx does not appear in Eq. (2). Please reconcile the definition with the physics of raster scanning.
  3. Since E_eff depends on ν, reproducibility requires reporting ν for each matrix/curve. I could not find a fixed or varied ν value in Sections 2–3, only the instrument’s capability range (20–200 kHz). Please add ν for every dataset in Figs. 5–10.
  4. Throughout the paper the CuBr laser is described as 30 ns, yet Figure 9’s caption says “CuBr laser with a pulse duration of 100 ns.” Please correct the caption or the methods.
  5. Contrast on copper is sensitive to surface finish, oxide film, contamination, and ambient gas. The methods say plates were cut to size, but do not describe cleaning (solvent/abrasive). Please report surface preparation, roughness before marking, and atmosphere during marking.
  6. Figure 4’s 3D views are used to claim “two times less roughness” at higher speed, yet no Ra/Rq/Rz values or confidence intervals are reported.
  7. Section 2.1 title is “Subsection. Please rename to a descriptive title.

Author Response

Comments 1: n Table 2, the density is given as ρ = 0.002 kg/m³, while the specific heat capacity is reported as c = 8940 J/(kg·K). These values are not physically plausible for copper and appear to be swapped (and/or units misassigned). Please correct the numbers and units.

Response 1: We have made the necessary correction - the correct Density ρ = 8.96 g/cm3. (line 110)

Comments 2: Definition of “effective energy” omits hatch spacing and is internally inconsistent with your text. You define as the absorbed energy per unit area, derived from linear energy density and linear pulse density. However, areal energy input for raster marking also depends on the line‑to‑line spacing Δx, as you rightly noted further E_eff “accounts for … speed and raster step,” but Δx does not appear in Eq. (2). Please reconcile the definition with the physics of raster scanning.

.

Response 2: Thank you for your comment. When planning the tasks and the purpose of our article, we were guided by the following considerations in choosing the physical complex quantity "effective energy":

   Effective energy is obtained from the product of the linear energy density and the linear pulse density.

  • The linear energy density (energy per unit length along the scan) is:

Elin =     [J/m]

where P is the average laser power (W) and v is the scanning speed (m/s).

  • The linear pulse density (number of pulses per unit length) is:

Dlin =    [pulses/s]

where  is the pulse repetition frequency (Hz).

  • Therefore, the effective energy can be expressed as:

Eeff = Elin · Dlin =   =      [J/m2]

   This parameter describes the energy delivered during the first transition of the laser beam across the surface, before any overlap between adjacent scan lines occurs. It reflects the initial conditions of the laser–matter interaction, taking into account laser power, speed, frequency, and pulse energy.

Areal energy

   The areal energy density is defined as:

Eareal =     [J/m2]   ,

     where raster step  is the hatch spacing between adjacent scan lines.

   This parameter inherently accounts for the step size between stripes. However, in practice, when processing with a small step (e.g. 30 – 90 % overlap), the absorption characteristics of the material change significantly. The irradiated zone has already been modified by prior pulses, leading to higher absorption and fundamentally new interaction dynamics compared to the initial pass.

Comments 3: Since E_eff depends on ν, reproducibility requires reporting ν for each matrix/curve. I could not find a fixed or varied ν value in Sections 2–3, only the instrument’s capability range (20–200 kHz). Please add ν for every dataset in Figs. 5–10.

.”

Response 3: Thank you for your comment. We have added the necessary clarification in Section 2.4 (Methodology) of the article: “In all experiments conducted on the four matrices, the frequency was kept constant at 20 kHz for both types of lasers. This clarification has been added to the methodology.” (line 148).

Comments 4: Throughout the paper the CuBr laser is described as 30 ns, yet Figure 9s caption says CuBr laser with a pulse duration of 100 ns. Please correct the caption or the methods.

Response 4: Thank you for your comment. We made a technical error. We have fixed this error with "pulse duration of 30 ns" (line 347).

Comments 5: Contrast on copper is sensitive to surface finish, oxide film, contamination, and ambient gas. The methods say plates were cut to size, but do not describe cleaning (solvent/abrasive). Please report surface preparation, roughness before marking, and atmosphere during marking.

Response5: Thanks for the identified gap in the methodology, the method for re-preparation of semples has been added. "All samples before laser processing are cleaned of the thin protective layer of oil by immersion in an ultrasonic bath with isopropanol for 45 minutes." (lines 154 - 156)

Comments 6: Figure 4s 3D views are used to claim two times less roughness at higher speed, yet no Ra/Rq/Rz values or confidence intervals are reported.

Response 6: Thank you for identifying the inaccuracy on our part. The issue has been fixed by adding the following text "a) 10 mm/s, Rz = 150 ± 4 μm; b) 70 mm/s, Rz = 75 ± 4 μm" to Figure 4. (lines 210)

Comments 7: Section 2.1 title is “Subsection. Please rename to a descriptive title.

Response 7: Thank you for your comment. The necessary correction has been made in subsection 2.1 (line 99).

Reviewer 4 Report

Comments and Suggestions for Authors
  1. In Table 2, the values for density and specific heat capacity appear to have been inadvertently swapped or incorrectly assigned. This should be corrected.
  2. It is recommended that Figure 1 include block diagrams of the laser systems, in addition to the photos of the devices, to improve clarity and understanding.
  3. The caption of Figure 3 lacks clarity and should be revised for better readability and consistency. The current phrasing ("...and CuBr laser d") is grammatically unclear and may confuse the reader. I suggest restructuring the caption to clearly distinguish which matrices were marked with which laser type. For example: “Figure 3. Copper matrices marked with different lasers:a–c) Matrices 1–3 marked with fiber laser, d) Matrix 4 marked with CuBr laser.”
  4. The caption of Figure 5 should more clearly reflect all relevant experimental parameters. Specifically, the fact that the data represent results obtained using three different power densities is not mentioned in the current caption. I recommend revising the caption to include this important detail, as it significantly affects the interpretation of the results. For example, the caption could be modified to something like “Graphics of the dependence of contrast on speed for marking a Cu plate with a fiber laser (pulse duration: 100 ns) at three different power densities.”
  5. Figure 6 appears to have the same issue as Figure 5 and should be revised accordingly.
  6. There appears to be a discrepancy between the colors mentioned in the caption of Figure 8 and those shown in the actual graph. The caption refers to gray, orange, and blue lines, while the figure itself appears to use blue, green, and red. I recommend revising the caption and/or the figure to ensure consistency and avoid confusion.
  7. Figure 9 seems to exhibit the same issue identified in Figure 8 and may require a similar revision.
  8. It is recommended that the caption of Figure 9 be modified to specify that the results are presented for three different scanning speeds.
  9. There appears to be a mismatch between the color description provided in lines 340–341 for Figure 10 (gray and green) and the colors actually shown in the figure (green and blue). It is recommended that the authors check and update the explanation to ensure consistency.
  10. The explanation in lines 354–357 could benefit from greater clarity. The term 'sloping' is somewhat vague in this context, and the connection between the saturation effect, contrast behavior, and the marking mechanism (melting) could be more explicitly articulated. Clarifying these relationships would improve the reader's understanding. If melting is the dominant marking mechanism in this energy range, it may also be helpful to explain how this conclusion was drawn.
  11. The sentence stating, "A comparative analysis of the two wavelengths (1030 nm for the fiber laser and 578 & 511 nm for the CuBr laser) on the marking process was also performed" (lines 374-376), is somewhat misleading, as it refers to "two wavelengths" but actually mentions three distinct wavelengths. Since the CuBr laser emits the 511 nm and 578 nm wavelengths simultaneously, it would be clearer to rephrase this sentence to reflect that the comparison is between the fiber laser wavelength (1030 nm) and the dual-wavelength emission of the CuBr laser. For example: "A comparative analysis of the fiber laser at 1030 nm and the CuBr laser emitting simultaneously at 511 and 578 nm was performed." This will improve clarity and precision.
  12. It is recommended that the conclusion explicitly state which combination of the tested laser parameter values is optimal for specific applications. Providing such practical guidance would be valuable for laser marking operators and enhance the applicability of the study.
Comments on the Quality of English Language

The manuscript would benefit from a careful language revision to improve the clarity and fluency of the English text.

Author Response

Comments 1: In Table 2, the values for density and specific heat capacity appear to have been inadvertently swapped or incorrectly assigned. This should be corrected

Response 1: Thank you for pointing out the discrepancy. The parameters in Table 2 have been corrected. (line 110)

Comments 2: It is recommended that Figure 1 include block diagrams of the laser systems, in addition to the photos of the devices, to improve clarity and understanding.

Response 2: A block diagram of the laser systems has been added with the positions of the main modules, which are essentially identical in the two laser technology systems (Rofin Power Line F 20 Varia Yb-doped fiber laser and CuBr MOPA laser) with which the experiments were conducted: 1 – Laser source; 2 – Galvo motor in X and Y; 3 – Focusing lens; 4 – Sample/worktable (lines 120 -122)

Comments 3:  The caption of Figure 3 lacks clarity and should be revised for better readability and consistency. The current phrasing ("...and CuBr laser d") is grammatically unclear and may confuse the reader. I suggest restructuring the caption to clearly distinguish which matrices were marked with which laser type. For example: “Figure 3. Copper matrices marked with different lasers:a–c) Matrices 1–3 marked with fiber laser, d) Matrix 4 marked with CuBr laser.”

Response 3: Thank you for your comment. We have taken your suggestion into consideration. The text in Fig. 3 has been corrected. (lines 193 - 194)

Comments 4: The caption of Figure 5 should more clearly reflect all relevant experimental parameters. Specifically, the fact that the data represent results obtained using three different power densities is not mentioned in the current caption. I recommend revising the caption to include this important detail, as it significantly affects the interpretation of the results. For example, the caption could be modified to something like “Graphics of the dependence of contrast on speed for marking a Cu plate with a fiber laser (pulse duration: 100 ns) at three different power densities.”.

Response 4: Thank you for your comment. In Fig. 5, according to your suggestion, a correlation has been made that reflects all the relevant experimental parameters in the text below the figure. (lines 233 - 235)

Comments 5: Figure 6 appears to have the same issue as Figure 5 and should be revised accordingly

Response5: Thank you for identifying the inaccuracy on our part. The problem has been fixed, the colors have been corrected in the text according to the colors in Figure 6. (lines 261 – 263)

Comments 6: There appears to be a discrepancy between the colors mentioned in the caption of Figure 8 and those shown in the actual graph. The caption refers to gray, orange, and blue lines, while the figure itself appears to use blue, green, and red. I recommend revising the caption and/or the figure to ensure consistency and avoid confusion.

Response 6: Thank you for identifying the inaccuracy on our part. The problem has been fixed, the colors have been corrected in the text according to the colors in Figure 8. (lines 309 - 311)

Comments 7: Figure 9 seems to exhibit the same issue identified in Figure 8 and may require a similar revision.

Response 7: Thank you for identifying the inaccuracy on our part. The problem has been fixed, the colors have been corrected in the text according to the colors in Figure 9. (lines 318 -319)

Comments 8: It is recommended that the caption of Figure 9 be modified to specify that the results are presented for three different scanning speeds.

Response 8: Thank you for your comment on Fig. 9 and your suggestion. We accept it and a correction has been made that reflects all relevant experimental parameters in the text below the figure. (lines 346 -348)

Comments 9: There appears to be a mismatch between the color description provided in lines 340–341 for Figure 10 (gray and green) and the colors actually shown in the figure (green and blue). It is recommended that the authors check and update the explanation to ensure consistency.

Response 9: Thank you for identifying the inaccuracy on our part. The problem has been fixed, the colors have been corrected in the text according to the colors in Figure 10. (lines 351 - 352)

Comments 10: The explanation in lines 354–357 could benefit from greater clarity. The term 'sloping' is somewhat vague in this context, and the connection between the saturation effect, contrast behavior, and the marking mechanism (melting) could be more explicitly articulated. Clarifying these relationships would improve the reader's understanding. If melting is the dominant marking mechanism in this energy range, it may also be helpful to explain how this conclusion was drawn.

Response 10: Thank you for the remark. After the correction, the text inserted in the article is as follows:For the effective energy interval of 15.52 kJ/cm² to 43.1 kJ/cm², the rate of contrast increase is very slow for both lasers, suggesting a saturation effect where additional energy has a negligible effect on contrast. In this interval, marking is mainly by melting, with only minor increases in marking depth.“ (lines 365 - 368)

Comments 11: The sentence stating, "A comparative analysis of the two wavelengths (1030 nm for the fiber laser and 578 & 511 nm for the CuBr laser) on the marking process was also performed" (lines 374-376), is somewhat misleading, as it refers to "two wavelengths" but actually mentions three distinct wavelengths. Since the CuBr laser emits the 511 nm and 578 nm wavelengths simultaneously, it would be clearer to rephrase this sentence to reflect that the comparison is between the fiber laser wavelength (1030 nm) and the dual-wavelength emission of the CuBr laser. For example: "A comparative analysis of the fiber laser at 1030 nm and the CuBr laser emitting simultaneously at 511 and 578 nm was performed." This will improve clarity and precision.

Response 11: Thank you for your comment. The conclusion has been substantially revised to take your recommendations into account. (See Conclusion from lines 377 to 403)

(lines 377 - 403)

Comments 12: It is recommended that the conclusion explicitly state which combination of the tested laser parameter values is optimal for specific applications. Providing such practical guidance would be valuable for laser marking operators and enhance the applicability of the study.

Response 12: Thank you for your comment. In the conclusion, we have indicated recommended effective energy intervals for each of the two technological lasers that show the best contrast and are recommended for successful work with both systems. “This study represents the first direct comparative analysis of Yb-doped fiber and CuBr MOPA lasers for copper marking, employing a multiparameter optimization approach. It defines optimal effective energy ranges: 17.4 - 43.1 kJ/cm² for the fiber laser and 9.90 - 43.1 kJ/cm² for the CuBr laser, providing practical, industry-relevant guidelines. This framework can be applied to future research on high-contrast laser marking.” (lines 399 - 403)

Round 2

Reviewer 1 Report

Comments and Suggestions for Authors

The authors responded to all comments and made the necessary edits to the manuscript.

Reviewer 2 Report

Comments and Suggestions for Authors

No comments

Reviewer 3 Report

Comments and Suggestions for Authors

The authors did an excellent job, and I believe the article can be accepted in its current form.

Reviewer 4 Report

Comments and Suggestions for Authors

I would like to thank the authors for their thorough and thoughtful revision of the manuscript. All of my comments and suggestions have been fully addressed, and the necessary corrections have been made appropriately. The revised version represents a clear improvement and now meets the scientific and methodological standards expected for publication in this journal.

I have no further comments, and I recommend the manuscript for publication in its current form.